# Read Operation Mechanism of Feedback Field-Effect Transistors with Quasi-Nonvolatile Memory States

**DOI:** 10.3390/nano14020210

**Published:** 2024-01-18

**Authors:** Juhee Jeon, Kyoungah Cho, Sangsig Kim

**Affiliations:** Department of Electrical Engineering, Korea University, 145 Anam-ro, Seongbuk-gu, Seoul 02841, Republic of Korea; isdf35@korea.ac.kr

**Keywords:** quasi-nonvolatile memory, TCAD, FBFET, read operation, rising time

## Abstract

In this study, the read operation of feedback field-effect transistors (FBFETs) with quasi-nonvolatile memory states was analyzed using a device simulator. For FBFETs, write pulses of 40 ns formed potential barriers in their channels, and charge carriers were accumulated (depleted) in these channels, generating the memory state “State 1 (State 0)”. Read pulses of 40 ns read these states with a retention time of 3 s, and the potential barrier formation and carrier accumulation were influenced by these read pulses. The potential barriers were analyzed, using junction voltage and current density to explore the memory states. Moreover, FBFETs exhibited nondestructive readout characteristics during the read operation, which depended on the read voltage and pulse width.

## 1. Introduction

Quasi-nonvolatile memory has been developed as a novel memory technology. Its retention time ranges from 64 ms to 10 years, which is the gap between volatile and nonvolatile memories [1]. The ability to store data without a power supply reduces power consumption [2,3]. This type of memory addresses the timescale gap in the memory hierarchy. Traditional memory has evolved to satisfy the demands of high-performance computing [4,5]. Volatile memory, such as static or dynamic random-access memory, offers an ultrahigh speed with a limited retention time [6,7,8]. Nonvolatile memory, such as NAND flash memory, provides a retention time of 10 years but exhibits a low operation speed [6,9]. With the current data-centric trend, a timescale gap in the memory hierarchy worsens memory bottlenecks [10,11]. Recent studies have demonstrated the potential of quasi-nonvolatile memory; however, improvements are still required [12,13]. To replace traditional memory successfully, the new memory must possess unlimited write/read cycles. Feedback field-effect transistors (FBFETs) have emerged as attractive candidates for quasi-nonvolatile memory owing to their fast write/read speeds, long retention times, and nondestructive readout characteristics [3]. Furthermore, their inherent operating mechanism ensures high endurance, and their compatibility with complementary metal–oxide–semiconductor (CMOS) technology indicates that they can potentially replace traditional memory fully [3,14].

In FBFETs, the positive feedback mechanism enables quasi-nonvolatile memory operations. Their memory states are defined by the presence (State 1) and absence (State 0) of excess charge carriers in the channels. When the potential barriers collapse (form) abruptly due to the accumulation (depletion) of excess charge carriers, the positive feedback loop is activated (eliminated) in their channels, and their memory state becomes State 1 (State 0) [15,16]. Consequently, State 1 (State 0) is perceived by reading the high (low) current. The speed and pulse configuration of the reading influence the formation of the potential barrier [17,18,19]; thus, the readout characteristics depend on the read pulse. Understanding the read mechanism is crucial for further enhancing quasi-nonvolatile characteristics. In this study, the read operation of FBFETs with quasi-nonvolatile memory states was analyzed using technology computer-aided design (TCAD) simulation. The read operation of the FBFETs was newly interpreted via the analysis of a positive feedback mechanism at a nanosecond-order speed. Junction voltage and current density analysis were employed to examine the carrier type predominantly influencing the formation of the positive feedback loop and the role of channel potential barriers during the read operation. This study provides valuable insight into optimizing the FBFET design for performance enhancement and suggests the potential for achieving ultra-high-speed operation through a new reading scheme.

## 2. Results and Discussions

Figure 1a shows a schematic of a representative FBFET with a p-n-p-n structure. The dimensional parameters were a gated channel length (*L*_G_) of 50 nm, a nongated channel length (*L*_NG_) of 50 nm, a channel thickness (*t*_Si_) of 20 nm, and a gate oxide SiO_2_ thickness (*t*_ox_) of 5 nm. The doping concentrations were 3 × 10^19^ cm^−3^ for the source and drain regions and 5 × 10^18^ cm^−3^ for the gated and nongated channel regions. The gate, drain, and source electrodes were made of aluminum (work function = 4.0 eV). These parameters are summarized in Table 1. Figure 1b shows a schematic of a *V*_DS_ pulse (with *V*_GS_ = 0 V) for reading quasi-nonvolatile memory states. During the read operation, *V*_DS_ increased to the reading voltage (*V*_Read_) within the rising time (*T*_rise_) and fell to 0 V during the falling time (*T*_fall_), the *T*_fall_ value was the same as the *T*_rise_ value. Memory states were verified by measuring the drain-to-source current (*I*_DS_) during the reading time (*T*_Read_). The quasi-nonvolatile memory operation is depicted in Figure 1c, where the *T*_rise_ and *T*_Read_ values were 10 and 20 ns, respectively. The sequence Write 1-Read-Write 0-Read was performed to verify the memory states. After the write operation, the *I*_DS_ values were ~12 μA and 3.8 pA for States 1 and 0, respectively. Moreover, the retention time was 3 s, as observed in the full memory cycle (Figure 1d).

Figure 2 shows the output characteristics (a) and band diagram (b) of the FBFETs. For the *T*_rise_ values of 1, 10^−3^, and 10^−8^ s, the FBFETs exhibited bistable characteristics (in the forward and reverse drain-to-source voltage (*V*_DS_) sweeps) caused by the positive feedback loop associated with the potential barrier and carrier accumulation in the channel. A positive feedback loop was activated when the repeated accumulation of excess carriers collapsed the potential barrier in the channel. For a *T*_rise_ of 1 s, the *I*_DS_ abruptly increased at a *V*_DS_ of 1.47 V in the sweep of *V*_DS_ from 0 to 2.0 V, which was the latch-up phenomenon. For a *T*_rise_ shorter than 1 s, the formation of the potential barrier was hindered sufficiently to block the carrier flow; for a *T*_rise_ of 10^−8^ s, the potential barrier in the gated channel decreased as *V*_DS_ increased (Figure 2b). Consequently, the latch-up voltage decreased from 1.47 to 1.26 V and the transient leakage current increased from ~10^−16^ to ~10^−9^ A as *T*_rise_ decreased from 1 to 10^−8^ s.

The formation of the potential barrier affects each p-n junction of the FBFETs during the read operation. As shown in Figure 3a, the junction voltages (V_1_, V_2_, and V_3_) are defined as the differences between the electron and hole quasi-Fermi levels (E_FN_ and E_FP_, respectively). Note that the positive V_2_ was the reverse bias to the center junction; the equation *V*_Read_ = V_1_ + V_2_ + V_3_ was established before the latch-up [20]. The junction voltages indicated the potential barrier heights of the channels. The accumulation (depletion) of charge carriers in the channels generated a memory state: State 1 (State 0). The accumulation and depletion of charge carriers are described in detail in the Appendix A. The potential barrier in the gated channel was higher by 0.4 eV in State 0 than in State 1 (Figure 3b). Thus, at *V*_Read_ = 0 V, the initial values of V_1_ and V_2_ differed by 0.40 V between States 1 and 0, which corresponded to the difference in the potential barrier in the gated channel. As *V*_Read_ w\s swept for a *T*_rise_ of 10 ns (Figure 3c), V_1_ and V_2_ sustained the difference between States 1 and 0 during the *V*_Read_ sweep, whereas V_3_ was the same for States 1 and 0 before latch-up. The sustained potential barrier induced a difference between States 1 and 0. As *V*_Read_ reached 1.05 V, V_1_ and V_3_ increased, and V_2_ decreased abruptly in State 1. These results indicate that the potential barrier collapsed at 1.05 V in State 1 but not in State 0. V_1_ accounted for most of *V*_Read_ in both states. An increase in V_1_ signified an increase in the forward bias applied to the drain-side p-n junction. Therefore, the collapse of the channel potential barrier in State 1 arose from the unintended carrier flow at the drain-side p-n junction.

When a positive *V*_DS_ was applied to the FBFETs, electrons flowed toward the drain, whereas holes flowed toward the source. Some electrons injected from the source accumulated in the potential well, whereas the rest passed through the channel. Thus, the electron current density (*J*_n_) at the drain-side p-n junction represents the electrons that pass through, which are related to the increase in the V_1_ value. In contrast, the hole current density (*J*_p_) at the drain-side p-n junction refers to hole injection from the drain. The number of injected holes is equal to the sum of the numbers of holes that are accumulated, recombined (or generated), and passed through. *J*_n_ and *J*_p_ at the drain-side p-n junction are plotted as functions of *V*_DS_ in Figure 4. For a *T*_rise_ of 10 ns, as *V*_Read_ increased to 2.00 V, *J*_n_ instantly increased in State 1, whereas it remained at 10^−11^ A/cm in State 0 until a *V*_Read_ of 0.42 V (Figure 4a). However, *J*_p_ remained at ~10^2^ A/cm before the latch-up in both memory states (Figure 4b). As *J*_n_ reached the same magnitude as *J*_p_ (~10^2^ A/cm), *J*_p_ increased and eventually led to a latch-up; the increase in *J*_n_ induced the collapse of the channel potential barrier. As the *J*_n_ curve in State 0 was shifted rightward by 0.42 V compared with State 1, the latch-up phenomenon occurred at *V*_Read_ = 1.05 V for State 1, whereas it did not for State 0. The results showed that the flow of electrons in the channel influenced the activation of the positive feedback loop. In State 0, the higher potential barrier in the gated channel inhibited the flow of electrons, resulting in a difference from State 1.

Figure 5 shows the *I*_DS_ response to the read operation for various *V*_Read_ values in States 1 and 0 with a *T*_rise_ of 10 ns. The *I*_DS_ value indicates the memory state during *T*_Read_. Accordingly, the moderate *V*_Read_ range depends on the selection of *T*_Read_. For *T*_Read_ = 20 ns, marked with dotted lines in the figure, States 1 and 0 exhibited distinct *I*_DS_ results during the read operation. As shown in Figure 5a, as *V*_Read_ decreased to 0.90 V, the positive feedback loop activation was delayed in State 1. However, even if *V*_Read_ was less than the latch-up voltage, the positive feedback loop was activated in State 1. In Figure 5b, for State 0, the *I*_DS_ values stabilized at ~10^−16^ A over time. The carrier flows were hindered by the potential barrier in State 0 during *V*_Read_; thus, the FBFETs exhibited nondestructive readout characteristics.

An investigation of the read operation is necessary to improve the reading speed of quasi-nonvolatile memory states. The memory window, namely, the *V*_Read_ range for recognizing memory states, is not exclusively defined by the DC characteristics of the FBFETs. At nanosecond-order speeds, the flow of charge carriers in the channel influences the activation of a positive feedback loop. To enhance the current-sensing margin, the reinforcement of the gated channel barrier is required to suppress the carriers passing through and reduce the leakage current. In State 1, charge carriers are accumulated in the gated channel after the read operation, which reduces the activation time of the positive feedback loop for the subsequent read operation. In State 0, owing to the high-potential barrier, charge carriers remain depleted in the gated channel after the read operation (for more details, see the Appendix A). The nondestructive readout characteristics enable stable and recursive read operations [21]. Therefore, incorporating the initialized operation into the read pulse scheme allows FBFETs to operate at an SRAM-like speed (sub-1 ns). Moreover, applying a flexible pulse scheme during repetitive read operations can improve the performance of FBFETs. Consequently, a suitable tool can be designed for a specific purpose through an in-depth understanding of the reading operation of FBFETs.

## 3. Conclusions

The read operations of FBFETs were investigated with quasi-nonvolatile memory states. FBFETs exhibited a retention time of 3 s for write and read pulses of 40 ns. State 1 (State 0) was generated by the accumulation (depletion) of charge carriers in the channels. During the rise in *V*_Read_ at a *T*_rise_ of 10 ns, the increase in electron flow at the drain-side p-n junction affected the activation of the positive feedback loop in the channel. In State 1, as *J*_n_ reached the same magnitude as *J*_p_ (~10^2^ A/cm), the potential barrier collapsed at a *V*_Read_ of 1.05 V, which caused a latch-up. In State 0, the high-potential barrier in the gated channel inhibited the flow of electrons, which differed from State 1. Consequently, the read operation was distinguishable for States 1 and 0 with nondestructive characteristics. Moreover, the range of *V*_Read_ depended on the pulse width during the read operation. The results provided insights for improving the performance of FBFETs as quasi-nonvolatile memory devices.

## 4. Simulation Method

The simulation was performed with a two-dimensional structure using the commercial device simulator Synopsys Sentaurus (O_2018.06) [22]. The Lombardi, Philips unified mobility, and high-field saturation models were used to consider the doping and field dependencies of carrier mobility. The Fermi statistics were used to perform an accurate simulation. Also, bandgap narrowing (Slotboom model), band-to-band tunneling, Shockley–Read–Hall (SRH) recombination with concentration-dependent lifetimes, surface SRH recombination, and Auger recombination were considered. Furthermore, an area factor of 20 nm was specified in the simulation to determine the device width.

## Figures and Tables

**Figure 1 nanomaterials-14-00210-f001:**
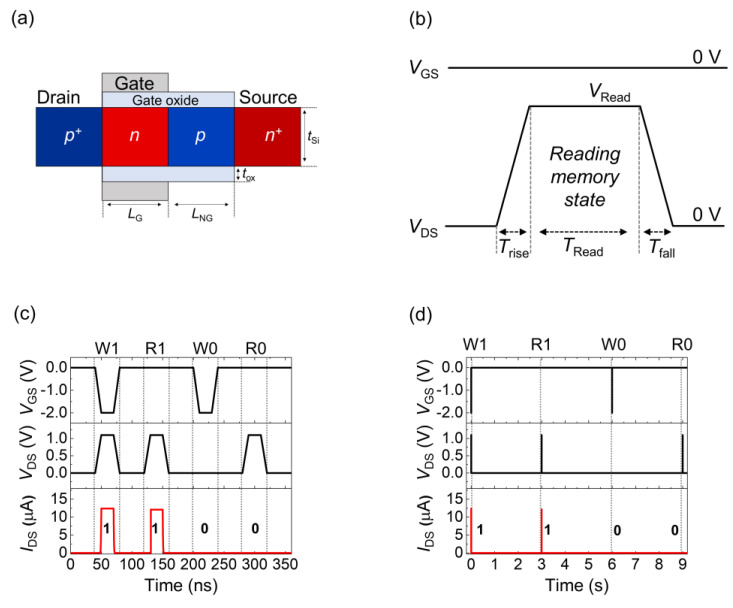
Schematic of (**a**) an FBFET and (**b**) read operation condition. Timing diagrams of (**c**) quasi-nonvolatile memory operation and (**d**) retention.

**Figure 2 nanomaterials-14-00210-f002:**
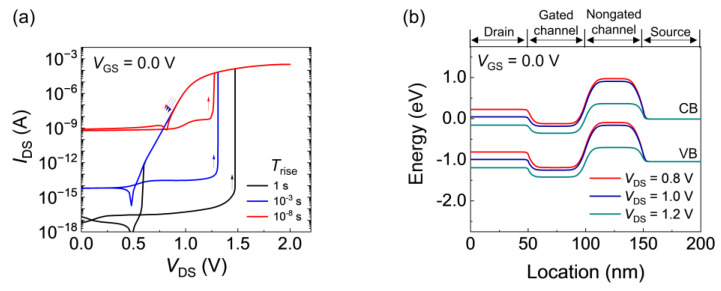
(**a**) *I*_DS_–*V*_DS_ curves depending on *T*_rise_. (**b**) Energy band diagrams at various drain voltages at a *T*_rise_ of 10 ns.

**Figure 3 nanomaterials-14-00210-f003:**
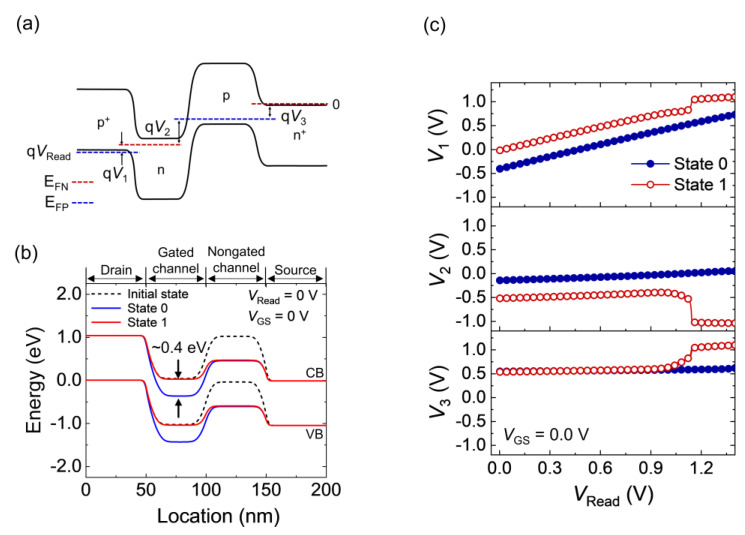
Energy band diagrams for (**a**) junction voltages and (**b**) memory states. (**c**) Junction voltages versus *V*_DS_ for States 1 and 0 at a *T*_rise_ of 10 ns.

**Figure 4 nanomaterials-14-00210-f004:**
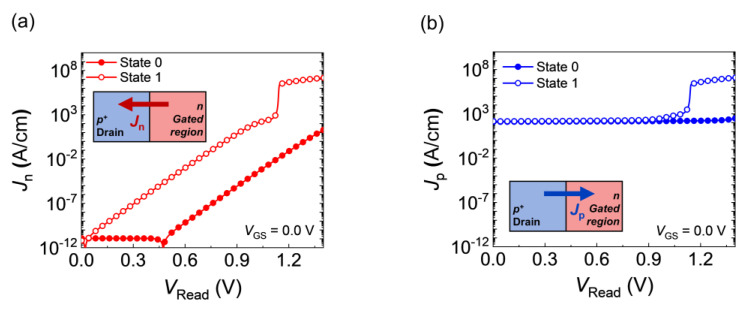
(**a**) Electron and (**b**) hole current densities at the drain-side p-n junction for States 1 and 0 at a *T*_rise_ of 10 ns.

**Figure 5 nanomaterials-14-00210-f005:**
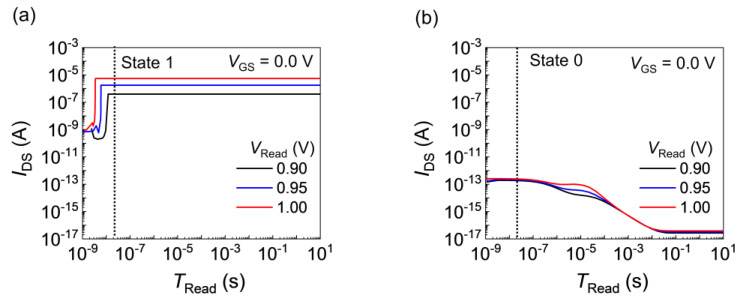
*I*_DS_ response as a function of *T*_Read_ at a *T*_rise_ of 10 ns for (**a**) State 1 and (**b**) State 0. The *I*_DS_ response depends on *V*_Read_.

**Table 1 nanomaterials-14-00210-t001:** Dimensional parameters and doping concentrations for simulations.

Simulation Parameters	Value (Unit)
Drain/source doping concentration	3 × 10^19^ (cm^−3^)
Gated/nongated channel doing concentration	5 × 10^18^ (cm^−3^)
Gated channel length (*L*_G_)	50 (nm)
Nongated channel length (*L*_NG_)	50 (nm)
Channel thickness (*t*_Si_)	20 (nm)
Gate oxide thickness (*t*_ox_)	5 (nm)
Work function of gate, drain, and source electrodes	4.0 (eV)

## Data Availability

Data are contained within the article.

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
