# Peer review of "Read Operation Mechanism of Feedback Field-Effect Transistors with Quasi-Nonvolatile Memory States"

_nanomaterials, 2024, doi:10.3390/nano14020210_

Round 1

Reviewer 1 Report

Comments and Suggestions for Authors

In this research, the author discusses the read operation mechanism of feedback field-effect Transistors with quasi-nonvolatile memory states by TCAD. From their results, write pulses of 40 ns form potential barriers in their channels, and charge carriers are accumulated (depleted) in these channels, generating the memory state “State 1 (State 0)”. Read pulses of 40 ns read these states with a retention time of 3s, and the read pulses influence the potential barrier formation and carrier accumulation. The potential barriers were analyzed using junction voltage and current density to explore the memory states.  However, here are a few questions for this research.

1.     For the charge carriers are accumulated (depleted) in these channels issue, please showing the charge carriers simulated distribution in a 2D or 3D image.

2.     For the potential barrier formation and carrier accumulation influenced by the read pulses, please give the simulated carriers' distribution before and after the read pulse in a 2D or 3D image.

3.     There are many simulation parameters within the TCAD. The author should list them in a table. This could help the reader to simulate the results. 

Reviewer 2 Report

Comments and Suggestions for Authors

The following issues must be addressed:

1.       Remove all pronouns (We, our,…) from the manuscript;

2.       Introduction part should be significantly improved by outlining what is new and innovative in this work;

3.       Figures resolutions must be improved;

4.       Discussion part should not resume on presenting the results that can be observed from different graphs; the author must focus on comparing the results and providing insight perspectives.

5.       Conclusion part is too general as well. Must focus on the most representative results.

Round 2

Reviewer 2 Report

Comments and Suggestions for Authors

The manuscript can be published in present form.